# Adaptive Budget Allocation for Orthogonal Subspace Adapter Tuning in LLMs Continual Learning

**Zhiyi Wan**[*]                                                                      *wzy10@bupt.edu.cn*
*State Key Laboratory of Networking and Switching Technology*
*Beijing University of Posts and Telecommunications*

**Wanrou Du**[*]                                                                  *wanroudu@bupt.edu.cn*
*State Key Laboratory of Networking and Switching Technology*
*Beijing University of Posts and Telecommunications*

**Yijia Chi**                                                                          *yijia@bupt.edu.cn*
*Beijing University of Posts and Telecommunications*

**Liang Li**                                                                              *lil03@pcl.ac.cn*
*Department of Strategic and Advanced Interdisciplinary Research*
*Pengcheng Laboratory*

**Miao Pan**                                                                             *mpan2@uh.edu*
*Department of Electrical and Computer Engineering*
*University of Houston*

**Xiaoqi Qin**                                                                      *xiaoqiqin@bupt.edu.cn*
*State Key Laboratory of Networking and Switching Technology*
*Beijing University of Posts and Telecommunications*

**Reviewed on OpenReview:** *https://openreview.net/forum?id=LNGDlLOdex*

## Abstract

Large language models (LLMs) often suffer from catastrophic forgetting in continual learning (CL) scenarios, where performance on previously learned tasks degrades severely while training on sequentially arriving tasks. Although pioneering CL approaches using orthogonal subspaces can mitigate task interference, they typically employ fixed budget allocation, neglecting the varying complexity across tasks and layers. Besides, recent budget-adaptive tuning methods for LLMs often adopt multi-stage paradigms that decouple optimization and budget allocation. Such decoupling results in potential misalignment, which hinders those approaches' practical application in CL scenarios. To address these limitations, we propose OA-Adapter, a novel parameter-efficient approach for continual learning in LLMs that unifies dynamic budget adaptation with orthogonal subspace learning in an end-to-end training stage. Specifically, OA-Adapter introduces a dynamic bottleneck dimension adaptation mechanism that simultaneously allocates an efficient parameter budget and optimizes task objectives without misalignment. To effectively preserve previously acquired knowledge while coordinating with the dynamic budget allocation, orthogonal constraints are applied specifically between the parameter subspace of the current task and the dynamically allocated parameter subspaces of historical tasks. Experimental results on continual learning benchmarks demonstrate that OA-Adapter outperforms state-of-the-art methods in both accuracy and parameter efficiency. OA-Adapter achieves higher average accuracy while using 58.5% fewer parameters on the Standard CL Benchmark, and maintains its advantages on two larger benchmarks comprising 15 tasks.

---

[*]Equal contribution.

# 1 Introduction

Recent advances in large language models (LLMs) have transformed artificial intelligence by demonstrating remarkable capabilities across diverse domains (Zhou et al., 2024a; Touvron et al., 2023). However, real-world deployment demands that LLMs continually adapt to evolving user needs and emerging tasks while retaining previously acquired knowledge—a prerequisite for sustainable lifelong learning (Xi et al., 2025; Wang et al., 2023b).

Parameter-efficient fine-tuning (PEFT) methods, such as adapter modules (Houlsby et al., 2019) and low-rank adaptation (LoRA) (Hu et al., 2022), enable task-specific adaptation by updating only $0.01\% - 4\%$ of model weights (Balne et al., 2024)). While originally designed to reduce computational costs for single-task tuning (Zhou et al., 2024b; Chen et al., 2024), these methods struggle in continual learning with sequentially arriving tasks. Sequentially tuning to arriving tasks induces catastrophic forgetting-severe degradation of performance on prior tasks (McCloskey & Cohen, 1989). One intuitive solution is to store task-specific adapters for each new task. However, this approach consumes substantial storage resources and leads to considerable inflexibility in multi-task deployments. Alternatively, retraining models with archived historical and additional new data necessitates frequent model updates and large data repositories (Shi et al., 2024). Both strategies are prohibitively costly and impractical in resource-constrained environments.

To efficiently adapt LLMs to downstream tasks while preserving previously acquired knowledge, researchers proposed continual learning (CL) methods. However, most CL approaches operate within shared parameter spaces across tasks, which inherently induces cross-task interference (Wang et al., 2024; Shi et al., 2024; Chaudhry et al., 2019; Shi & Wang, 2023; Rebuffi et al., 2017; Aljundi et al., 2018; Schwarz et al., 2018; Rongali et al., 2020; Lin et al., 2022). Furthermore, unlike conventional CL (e.g., incremental image classification) that typically handle tasks with limited distribution shifts , CL of LLMs often deals with substantially divergent task distributions, thus significantly amplifying interference when tuning in a shared parameter space. Some architecture-based methods further construct task-specific parameters to resolve this problem, but they heavily rely on explicit task ID during inference (Razdaibiedina et al., 2023; Jang et al., 2022; Jin et al., 2022; Li & Lee, 2024; Yan et al., 2023). Recent researches in orthogonal subspace learning (Farajtabar et al., 2020; Guo et al., 2022; Wang et al., 2023a) offer a promising alternative by restricting task-specific updates to mutually orthogonal parameter subspaces to eliminate interference, without task ID dependency during inference. However, existing methods typically rely on a fixed budget allocation, assigning the same subspace dimensionality to different task and layer. This rigid strategy overlooks the heterogeneity of task complexity and layer-specific adaptation needs, leading to inefficient parameter utilization, allocating excessive resource to simple tasks while under-allocating resources to more complex ones. Such inflexible allocation hinder LLMs' continuous adaptation capabilities in practice.

To achieve dynamic budget allocation, emerging budget-adaptive PEFT methods like AdaLoRA (Zhang et al., 2023), ElaLoRA (Chang et al., 2025), and DiffoRA (Jiang et al., 2025) proposed multi-stage paradigms with sequential optimization and budget adjustment phases. Such decoupled optimization may create misalignment between fine-tuning objectives and budget allocation. Moreover, the inherent complexity of multi-stage designs introduces substantial computational overhead and engineering challenges, limiting their practicality for continual learning systems.

We propose OA-Adapter to address these issues, a novel parameter-efficient approach for continual learning in LLMs. Instead of manually assigning a fixed budget, OA-Adapter automatically adjusts the parameter budget for each task and layer based on the task difficulty and model capacity.

Our key contributions are as follows:

- We propose OA-Adapter, a novel parameter-efficient approach for continual learning in LLMs that unifies dynamic budget adaptation with orthogonal subspace learning in an end-to-end training stage. To the best of our knowledge, this is the first work to integrate budget adaptation into parameter-efficient fine-tuning for continual learning in LLMs.

- We design a dynamic bottleneck dimension adaptation mechanism that simultaneously allocates an efficient parameter budget and optimizes task objectives without misalignment.

- We demonstrate that the parameter budget requirements in continual learning vary with task difficulty and position in the training sequence, and that the budget requirements of parameters in different layers vary, which emphasizes the necessity of our approach.

- Experimental results demonstrate that OA-Adapter outperforms state-of-the-art methods in both accuracy and parameter efficiency. OA-Adapter achieves higher average accuracy with 58.5% fewer parameters on the Standard CL Benchmark and maintains its advantages on two larger benchmarks comprising 15 tasks.

## 2 Related Work

**Continual Learning for LLMs.** Existing continual learning (CL) techniques for LLMs are mainly based on replay, regularization, and architecture modification, which have been most extensively applied in this context (Wang et al., 2024; Shi et al., 2024). Replay-based methods (Chaudhry et al., 2019; Shi & Wang, 2023; Rebuffi et al., 2017) store a subset of past data to rehearse on previous tasks during new training. However, this approach raises significant privacy issues and imposes high storage and computational costs, especially for LLMs. Regularization-based methods (Aljundi et al., 2018; Schwarz et al., 2018; Rongali et al., 2020; Lin et al., 2022) introduce a regularization term that penalizes significant weight deviations across tasks. However, their computational complexity grows rapidly with the number of tasks, and performance on previous tasks deteriorates significantly. These above approaches primarily focus on learning all incremental tasks within a shared parameter space, which is a major contributor to task interference. In contrast, many architecture-based methods (Razdaibiedina et al., 2023; Jang et al., 2022; Jin et al., 2022; Li & Lee, 2024; Yan et al., 2023) address this challenge by incorporating task-specific components, isolated parameters, or dedicated pathways within the model. They essentially learn task-specific expert modules for different tasks, so rely heavily on explicit task ID during inference. Recent advancements have explored a promising direction that can retain generalization capacity while reducing task interference without requiring explicit task ID. These methods (Farajtabar et al., 2020; Guo et al., 2022; Wang et al., 2023a) constrain weight updates for each task to lie within mutually orthogonal subspaces of the high-dimensional parameter space. By doing so, they effectively decouple the optimization directions of different tasks, thereby mitigating task interference. However, a common limitation of existing orthogonal subspace methods is the fixed parameter budget for all tasks and layers.

**Adaptive Budget Tuning.** Existing budget-adaptive tuning methods for LLMs commonly adopt a multi-stage design where budget adaptation is a separate stage from optimization. For example, DiffoRA (Jiang et al., 2025) trains a LoRA module for each layer in the first stage, and then selects a subset of these modules based on the "Difference-aware Adaptive Matrix" in a subsequent stage. ElaLoRA (Chang et al., 2025) involves a "Warm-up" fine-tuning stage, followed by a "Dynamic Rank Adjustment" stage to adapt LoRA module ranks, and finally a "Stabilization" fine-tuning stage. AdaLoRA (Zhang et al., 2023) injects rank pruning operations after iterations by applying SVD to selectively retain low-rank components. While these methods achieve strong performance on single-task datasets, their multi-stage design introduces substantial computational and engineering overhead that compromises practicality and scalability, while creating misalignment between optimization objectives and budget adaptation criteria that ultimately hinders the achievement of truly optimal solutions. And their design is restricted to unidirectional rank reduction, which does not allow for bidirectional budget adaptation. These limitations pose significant challenges when extending such methods to continual learning, which demands efficient, unified, and adaptive training processes for sequential tasks. Moreover, although recent methods (Liu et al., 2024; Chang et al., 2025; Ding et al., 2023) provide strong interpretability, they rely on task-specific heuristic configurations rather than adapting to the inherent characteristics of sequentially arriving tasks.

## 3 Methodology

In this section, we introduce OA-Adapter, a novel framework for continual learning in LLMs that simultaneously improves parameter efficiency and mitigates catastrophic forgetting in an end-to-end training stage, as

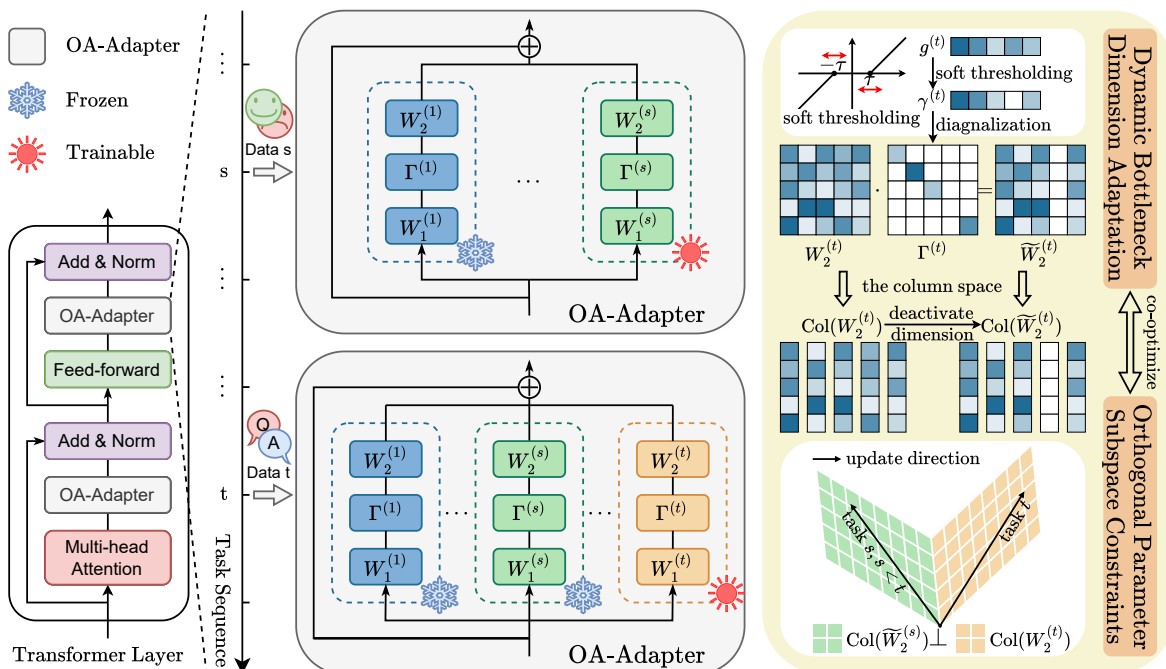

Figure 1: The OA-Adapter framework for LLM continual learning. Each task-specific OA-Adapter module (task $t$) comprises three core components: (1) a down-projection layer $\mathcal{W}_1^{(t)}$, (2) a diagonal mask $\Gamma^{(t)}$ with trainable threshold $\tau^{(t)}$, (3) and an up-projection layer $\mathcal{W}_2^{(t)}$. The dynamic masking mechanism enables bidirectional dimension adaptation through activation/deactivation of latent dimensions. Orthogonal subspace constraints are enforced between the column space of the $t$-th task parameters $\text{Col}(\mathcal{W}_2^{(t)})$ and the dynamically allocated parameter subspaces of historical tasks $\text{Col}(\widetilde{\mathcal{W}}_2^{(s)})$ (for $s < t$). Here, $\widetilde{\mathcal{W}}_2^{(s)}$ incorporates only the activated dimensions from the $s$-th task.

illustrated in Figure 1. We first describe its architectural design, including core components and computation flow. Then, we analyze the mathematical foundations of the dynamic bottleneck dimension adaptation, demonstrating how trainable thresholds enable bidirectional activation and deactivation of dimensions during an end-to-end training phase. Finally, we formalize the orthogonal parameter subspace constraints mechanism and explain how it works in concert with the dynamic bottleneck dimension adaptation to achieve parameter-efficient continual learning.

## 3.1 Module Structure

**Standard Adapter.** To enable efficient adaptation of pre-trained language models (PLMs) to downstream tasks, adapter modules inject lightweight trainable parameters into PLMs while keeping the original weights frozen. These modules employ a bottleneck architecture to minimize trainable parameters, consisting of a down-projection layer that reduces the original $d$-dimensional representation $x$ to a lower dimension $r$, a nonlinear activation function $f(\cdot)$, and an up-projection layer that restores the features to dimension $d$. The architecture ensures near-zero initialization of the projection layers while maintaining a skip connection to preserve the original features during initial training stages. For an input representation $x \in \mathbb{R}^d$, the output $y \in \mathbb{R}^d$ can be formalized as:

$$y = x + \mathcal{W}_2 \cdot f(\mathcal{W}_1 \cdot x + b_1) + b_2, \tag{1}$$

where $\mathcal{W}_1 \in \mathbb{R}^{r \times d}, \mathcal{W}_2 \in \mathbb{R}^{d \times r}$ denote the down-projection and up-projection matrices. $\mathbf{b}_1 \in \mathbb{R}^r$ and $\mathbf{b}_2 \in \mathbb{R}^d$ denote the bias terms, respectively, with bottleneck dimension $r \ll d$.

**OA-Adapter.** Building upon the standard Adapter's bottleneck architecture, OA-Adapter introduces two structural modifications: 1) the removal of bias terms in projection layers to create a bias-free parameter space containing only linear transformations, and 2) the replacement of static non-linear activations with a trainable diagonal masking matrix $\Gamma$ that dynamically adjusts the bottleneck dimension, as detailed in Section 3.2. These modifications enable the enforcement of orthogonal parameter subspace constraints, as detailed in Section 3.3, to mitigate cross-task interference and co-optimization of budget adaptation with continual learning in an end-to-end training phase. Specifically, the forward computation of OA-Adapter operates as follows:

$$y = x + \mathcal{W}_2 \cdot \Gamma \cdot \mathcal{W}_1 \cdot x, \tag{2}$$

where $\mathcal{W}_1 \in \mathbb{R}^{r_{\max} \times d}$ and $\mathcal{W}_2 \in \mathbb{R}^{d \times r_{\max}}$ denote the down-projection and up-projection matrices, respectively, and $\Gamma \in \mathbb{R}^{r_{\max} \times r_{\max}}$ is a trainable diagonal masking matrix. Here, $r_{\max} \ll d$ represents the pre-defined maximum bottleneck dimension.

## 3.2 Dynamic Bottleneck Dimension Adaptation

**Adaptation Mechanism.** To dynamically allocate parameter budget, we adjust the effective bottleneck dimensions of OA-Adapter using a trainable diagonal masking matrix $\Gamma \in \mathbb{R}^{r_{\max} \times r_{\max}} : \Gamma = \mathrm{diag}(\gamma)$. The sparsity of the vector $\gamma \in \mathbb{R}^{r_{\max}}$ is controlled via a soft thresholding mechanism applied to a trainable vector $g \in \mathbb{R}^{r_{\max}}$. Specifically, each diagonal entry $\gamma_i$ is computed as:

$$\gamma_i = \mathrm{soft}(g_i; \tau) = \mathrm{sign}(g_i) \cdot \max(|g_i| - \tau, 0), \tag{3}$$

where $\tau > 0$ is a trainable threshold that dynamically modulates the sparsity level of $\Gamma$ throughout the training process. The projection path of OA-Adapter can then be equivalently reformulated as:

$$\mathcal{W}_2 \cdot \Gamma \cdot \mathcal{W}_1 = \sum_{i=1}^{r_{\max}} \gamma_i \cdot \mathcal{W}_2[:, i] \otimes \mathcal{W}_1[i, :], \tag{4}$$

where $\otimes$ denotes the outer product. This decomposition clearly demonstrates how each $\gamma_i$ dynamically adjusts the contribution of the $i^{\mathrm{th}}$ latent dimension pair: when $|g_i| \leq \tau$, we have $\gamma_i = 0$, causing both the $i^{\mathrm{th}}$ column of $\mathcal{W}_2$ and $i^{\mathrm{th}}$ row of $\mathcal{W}_1$ to be disabled, effectively deactivating the corresponding dimension. This mechanism adaptively controls the bottleneck dimension through $r_{\mathrm{eff}} = \|\gamma\|_0$, where $\|\gamma\|_0$ represents the count of non-zero entries in $\gamma$.

**Gradient Analysis.** Our method's bidirectional dimension adaptation capability, enabled by the trainable threshold $\tau$, offers critical advantages. When $|g_i| \leq \tau$, the corresponding dimension pair is deactivated by setting $\gamma_i = 0$. This zeros the $i$-th diagonal entry of the masking matrix $\Gamma$, effectively removing that dimension's contribution in forward propagation. However, this operation also blocks gradient flow from the downstream loss $\mathcal{L}$ to $g_i$ when $\gamma_i = 0$. To clarify why $g_i$ becomes non-trainable in such cases, consider the gradient calculation via chain rule:

$$\frac{\partial \mathcal{L}}{\partial g_i} = \frac{\partial \mathcal{L}}{\partial \gamma_i} \frac{\partial \gamma_i}{\partial g_i}. \tag{5}$$

The derivative $\frac{\partial \gamma_i}{\partial g_i}$ of the soft thresholding function equals $\mathrm{sign}(g_i)$ when $|g_i| > \tau$, but critically becomes 0 when $|g_i| \leq \tau$. This implies that deactivated dimensions (where $|g_i| \leq \tau$) produce zero gradients in Equation (5) due to $\frac{\partial \gamma_i}{\partial g_i} = 0$, blocking gradient updates through $g_i$. With a fixed threshold $\tau$, such dimensions would remain permanently disabled throughout training. Crucially, our method implements $\tau$ as a learnable parameter shared across all dimensions within each OA-Adapter module. Thus, the gradient of $\tau$ with respect to the total loss $\mathcal{L}$ is:

$$\frac{\partial \mathcal{L}}{\partial \tau} = \sum_{i=1}^{r_{\max}} \frac{\partial \mathcal{L}}{\partial \gamma_i} \frac{\partial \gamma_i}{\partial \tau}, \tag{6}$$

where $\frac{\partial \gamma_i}{\partial \tau} = -\mathrm{sign}(g_i)$ when $|g_i| > \tau$ and 0 otherwise. This derivative relationship ensures threshold updates are primarily governed by dimensions exceeding the current $\tau$. As $\tau$ evolves during training, dimensions previously deactivated with $|g_i| \leq \tau$ may become reactivated when they satisfy $|g_i| > \tau$ under the updated

threshold, thereby reactivating their corresponding projection paths. This bidirectional adaptation mechanism automatically suppresses dimensions while maintaining their potential for reactivation in later training iterations. The bidirectional nature of this dynamic parameter budget adaptation approach ensures optimal parameter allocation that continuously adapts to the evolving requirements of sequential tasks in continual learning.

### 3.3 Orthogonal Parameter Subspace Constraints for Continual Learning

**Continual Learning Setup.** Continual learning focuses on incrementally acquiring knowledge from evolving data distributions of sequential tasks while mitigating catastrophic forgetting of previously acquired knowledge. Formally, models are trained on a sequential stream of tasks denoted as $\{D_1, D_2, \cdots, D_t\}$. Each task $D_t = (x_t^i, y_t^i)_{i=1}^{n_t}$ consists of input instances $x_t^i \in \mathcal{X}_t$ paired with corresponding labels $y_t^i \in \mathcal{Y}_t$, where $\mathcal{X}_t$ and $\mathcal{Y}_t$ represent the task-specific input and label spaces. During the training phase for task $D_t$, model parameters $\Phi$ are updated exclusively using data from $D_t$. The objective of continual learning can be formalized as optimizing:

$$\max_{\Phi} \sum_{t=1}^{T} \sum_{\{x_i^t, y_i^t\} \in \mathcal{D}_t} \log P_{\Phi}(y_t^i \mid x_t^i). \tag{7}$$

**Orthogonal Parameter Subspace Constraints.** Catastrophic forgetting arises when subsequent adaptations overwrite parameters critical for previous tasks. To mitigate this, we introduce orthogonality constraints that enforce parameter updates across tasks to occupy mutually independent subspaces. Let $\Delta\Phi_k^{(l)}$ represent the OA-Adapter's parameter at layer $l$ for the $k$-th task. We have:

$$\Delta\Phi_k^{(l)} = \mathcal{W}_2^{(k,l)} \cdot \Gamma^{(k,l)} \cdot \mathcal{W}_1^{(k,l)} = \widetilde{\mathcal{W}}_2^{(k,l)} \cdot \mathcal{W}_1^{(k,l)}. \tag{8}$$

Here, the columns of $\widetilde{\mathcal{W}}_2^{(k,l)}$ serve as orthogonal basis vectors spanning the parameter update subspace for the $k$-th task at layer $l$, while $\mathcal{W}_1^{(k,l)}$ determines how these basis vectors are combined. We therefore formally define the task-specific parameter subspace as the column space of $\widetilde{\mathcal{W}}_2^{(k,l)}$, which intrinsically aligns with the activated dimensions for the $k$-th task through the dimension-selective masking operation of $\Gamma^{(k,l)}$. Thus, we enforce strict orthogonality to new OA-Adapter parameters across sequential tasks for each layer, ensuring new task adaptations occupy parameter subspaces orthogonal to previous tasks' frozen parameter subspaces. Formally, the constraints for the $t$-th task at layer $l$ are defined as:

$$\langle \mathcal{W}_2^{(t,l)}[:,i], \widetilde{\mathcal{W}}_2^{(s,l)}[:,j] \rangle = 0, \ \forall i, \ j, \ s < t. \tag{9}$$

The columns of $\widetilde{\mathcal{W}}_2^{(t,l)}$ inherit directional properties from $\mathcal{W}_2^{(t,l)}$, ensuring orthogonal relationships persist regardless of dynamic dimension activation patterns. These asymmetric orthogonality constraints enable simultaneous optimization of dynamic bottleneck dimension adaptation and historical knowledge preservation. To formalize this approach, we incorporate an orthogonality regularization term into the optimization objective. Specifically, the pairwise orthogonality loss between current task $t$ and each historical task $s < t$ at layer $l$ is quantified as:

$$\mathcal{L}_{\text{orth}}^{(s,t,l)} = \sum_{i,j} \left\langle \mathcal{W}_2^{(t,l)}[:,i], \widetilde{\mathcal{W}}_2^{(s,l)}[:,j] \right\rangle^2. \tag{10}$$

Minimizing the loss term $\mathcal{L}_{\text{orth}}^{(s,t,l)}$ drives the inner product toward zero, enforcing parameter subspace orthogonality. The complete training objective, integrating both task-specific performance term $\mathcal{L}_{\text{task}}^{(t)}$ and orthogonality constraints with explicit summation over layers, is formulated as:

$$\mathcal{L}_{\text{total}} = \mathcal{L}_{\text{task}}^{(t)} + \lambda_{\text{orth}} \cdot \sum_{l=1}^{L} \sum_{s<t} \mathcal{L}_{\text{orth}}^{(s,t,l)}, \tag{11}$$

where $\lambda_{\text{orth}}$ is a hyperparameter controlling the strength of orthogonal regularization.

**The synergistic benefits of adaptive rank allocation and orthogonal subspace constraints.** In Equations equation 8 and equation 9, OA-Adapter deliberately enforces orthogonality on the effective up-projection $\mathcal{W}_2^{(k,l)}\Gamma^{(k,l)}$ rather than on $\Gamma^{(k,l)}\mathcal{W}_1^{(k,l)}$. This choice establishes task-specific adaptations in mutually orthogonal subspaces, enabling efficient adaptive rank allocation while maximizing parameter utilization.

To understand the intuition, first consider a standard low-rank adapter without gating at layer $l$. The output is $y = \mathcal{W}_2^{(l)}(\mathcal{W}_1^{(l)}x)$, or $y = \sum_{i=1}^{r} \alpha_i w_{2,i}^{(l)}$, where the columns of $\mathcal{W}_2^{(l)}$ define the basis directions injected into the hidden state. Thus, the up-projection $\mathcal{W}_2^{(l)}$ determines the actual directions added to the representation. With task-specific diagonal gating $\Gamma^{(k,l)}$, OA-Adapter selectively activates a subset of these directions. By constraining orthogonality on $\mathcal{W}_2^{(k,l)}\Gamma^{(k,l)}$, we ensure different tasks occupy orthogonal subspaces in the high-dimensional hidden state space, preventing subsequent tasks from overwriting learned directions.

This combination yields critical synergistic benefits. The adaptive gating mechanism significantly simplifies the computation: since only active dimensions in $\Gamma^{(k,l)}$ contribute to the effective up-projection, the constraint is enforced over a reduced effective rank, lowering computational costs. More importantly, earlier tasks typically converge to lower effective ranks, leaving more of the total parameter space available for subsequent tasks. Enforcing orthogonality on the alternative $\Gamma^{(k,l)}\mathcal{W}_1^{(k,l)}$ would fail, as subsequent multiplication by $\mathcal{W}_2^{(k,l)}$ would violate orthogonality in the injected directions, reintroducing interference.

## 4 Experiments and Analysis

### 4.1 Experimental Settings.

**Datasets.** We evaluate OA-Adapter using three CL benchmarks for LLMs: 1) **Standard CL Benchmark** (Zhang et al., 2015): a continual learning benchmark comprising five text classification datasets: AG News, Amazon Reviews, Yelp Reviews, DBpedia, and Yahoo Answers. 2) **Long Sequence Benchmark** (Razdaibiedina et al., 2023): a continual learning benchmark of 15 classification datasets, including five tasks from the Standard CL Benchmark, four from the GLUE benchmark (MNLI, QQP, RTE, SST2) (Wang et al., 2019b), five from the SuperGLUE benchmark (WiC, CB, COPA, MultiRC, BoolQA) (Wang et al., 2019a), and the IMDB movie reviews dataset (Maas et al., 2011). 3) **SuperNI Benchmark** (Wang et al., 2022a): a benchmark of diverse NLP tasks with expert-written instructions, enabling rigorous benchmarking of practical continual learning settings for LLMs. Specifically, we focus on five task categories: dialogue generation, information extraction, question answering, summarization, and sentiment analysis. From each category, three tasks are selected, resulting in a sequence of 15 tasks. Following Razdaibiedina et al. (2023); Zhao et al. (2024), for all benchmarks we randomly select 1,000 samples per class for training. The task details and training sequences of tasks used in our experiments are provided in Appendix A.2.

**Metrics.** Let $a_{i,j}$ denote the test accuracy on the $i$-th task after training on the $j$-th task, the metrics for evaluating are: **(i) Average Accuracy(AA)** (Chaudhry et al., 2018) over all tasks after completing training on the final task, i.e., $\text{AA}_T = \frac{1}{T}\sum_{i=1}^{T} a_{i,T}$. **(ii) Forgetting Rate (F.Ra)** (Chaudhry et al., 2018) measures how much knowledge has been forgotten across the first $T-1$ tasks, i.e., $FR_T = \frac{1}{T-1}\sum_{t=1}^{T-1}\left(\max_{k\in\{1,\dots,T-1\}} a_{k,t} - a_{T,t}\right)$. **(iii) Backward Transfer (BWT)** (Ke & Liu, 2022) which measures how much the learning of subsequent tasks influences the performance of a learned task, i.e., $\text{BWT}_T = \frac{1}{T-1}\sum_{t=1}^{T-1}\left(a_{T,t} - a_{t,t}\right)$

**Baselines.** We compare our method against various CL baseline approaches, including: **Replay** finetunes the whole model with a memory buffer, and replay samples from old tasks when learning new tasks to avoid forgetting. **L2P (Wang et al., 2022b)** uses the input to dynamically select and update prompts from the prompt pool in an instance-wise fashion. **LFPT5 (Qin & Joty, 2021)** continuously train a soft prompt that simultaneously learns to solve the tasks and generate training samples, which are subsequently used in experience replay. **O-LoRA (Wang et al., 2023a)** train new LoRA parameters on a sequential series of tasks in orthogonal subspace while fixing the LoRA matrices of previous tasks. **ProgPrompt (Razdaibiedina et al., 2023)** adopts a task-specific soft prompt for each distinct task, sequentially appending it to prior learned prompts. In essence, it trains individual models per task, leveraging the task ID to select

Table 1: Testing performance on three benchmarks with T5-large, ↑ indicates higher values are better and ↓ indicates lower values are better.

| | Standard CL Benchmark | | | Long Sequence Benchmark | | | SuperNI Benchmark | | |
|---|---|---|---|---|---|---|---|---|---|
| | AA↑ | F.Ra↓ | BWT↑ | AA↑ | F.Ra↓ | BWT↑ | AA↑ | F.Ra↓ | BWT↑ |
| Replay | $57.8_{\pm0.1}$ | 12.3 | -13.4 | $54.2_{\pm0.4}$ | 13.5 | -12.2 | $20.5_{\pm0.2}$ | 14.7 | -15.8 |
| L2P | $60.7_{\pm0.6}$ | 15.3 | -11.2 | $56.1_{\pm0.4}$ | 19.4 | -16.6 | $12.7_{\pm0.2}$ | 11.9 | -8.0 |
| LFPT5 | $72.6_{\pm0.9}$ | 9.1 | -8.3 | $68.6_{\pm0.3}$ | 12.7 | -12.8 | $26.7_{\pm0.5}$ | 15.0 | -14.5 |
| O-LoRA | $75.3_{\pm0.2}$ | 12.8 | -9.1 | $68.7_{\pm0.6}$ | 7.0 | -4.1 | $25.9_{\pm0.4}$ | 26.4 | -24.6 |
| ProgPrompt | $75.1_{\pm0.7}$ | 11.3 | -8.1 | $63.2_{\pm0.3}$ | 9.4 | -6.7 | $23.3_{\pm0.6}$ | 13.7 | -13.2 |
| OA-Adapter | $\mathbf{76.0_{\pm0.4}}$**8.7** | | **-7.5** | $\mathbf{69.2_{\pm0.3}}$ | **4.4** | **-3.2** | $\mathbf{29.3_{\pm0.6}}$**8.2** | | **-6.0** |

the appropriate model during inference. In our continual learning baseline selection, we specifically focused on methods that could be reliably reproduced to ensure fair comparison. Furthermore, to guarantee the authenticity and consistency of our experimental results, we reproduced baselines in our infrastructure.

**Implementation Details.** All our experiments involving T5 models were performed on a server outfitted with four NVIDIA GeForce RTX 3090 GPUs, utilizing the DeepSpeed repository for implementation. Following previous studies (de Masson d'Autume et al., 2019; Rao et al., 2019), for each dataset, we use the available validation set as the test set(since test data is not available) and hold out 500 samples from the training set to construct the validation set. For every sequence of tasks across different orders, we trained the models for one epoch using a batch size of 32 (8 per GPU), a dropout rate of 0.1, and no weight decay. Across all experiments, we primarily used Adapter modules with a bottleneck dimension of 16, and applied a sparsification threshold chosen from $\{1e\text{-}3, 1e\text{-}4, 1e\text{-}5\}$. The learning rate was selected from $\{5e\text{-}3, 3e\text{-}3, 1e\text{-}3, 5e\text{-}4\}$ depending on task characteristics. We applied an orthogonality regularization on the Adapter's upsampling matrix with a coefficient $\lambda_{orth} \in \{0.5, 1, 5\}$, and used an additional coefficient $\lambda_2 \in \{0, 0.1, 0.5\}$ to scale the associated L2 loss term. To ensure experimental comparability, we maintain consistency with O-LoRA (Wang et al., 2023a) by adopting instruction tuning as the training paradigm. The specific instructions used for each task are detailed in Appendix A.3.

## 4.2 Main Results

Our experiments employ both the encoder-decoder T5 model (Raffel et al., 2020) and the decoder-only LLaMA-7B model (Touvron et al., 2023), consistent with baselines in CL for NLP. In Table 1, we present the main results using T5-large across three CL benchmarks. Following previous works (Qin & Joty, 2021; Wang et al., 2023a), we report the results of three independent runs with different task orders on each benchmark. All experimental results are reported as the average of three runs.

**Results on Continual Learning Benchmarks.** As illustrated in Table 1, across all task orders of the Standard CL Benchmark, OA-Adapter consistently surpasses previous methods by a significant margin. On the other two benchmarks comprising 15 tasks, OA-Adapter consistently outperforms previous methods. Notably, on the SuperNI dataset, which is primarily designed for generative tasks, CL methods still perform poorly, underscoring that continual learning for complex tasks remains a significant challenge. To disentangle the roles of orthogonal constraints and adaptive budget, we compare the full OA-Adapter with two variants: a no-orthogonality variant that removes the orthogonal parameter subspace constraint, and a fixed budget variant that disables dynamic bottleneck dimension adaptation. As shown in Table 2, removing orthogonality leads to significant degradation, underscoring its importance in mitigating catastrophic forgetting; disabling adaptation also reduces performance, indicating that adaptive capacity allocation provides additional gains.

**Parameter Efficiency Analysis.** As established in Section 3.2, OA-Adapter leverages dynamic parameter budget allocation to achieve enhanced parameter efficiency. We compare the parameter utilization between OA-Adapter and O-LoRA across various initial budget conditions, as illustrated in Table 3. Parameter budget represents bottleneck dimension distribution for OA-Adapter, while determines the module's

Table 2: Ablation study to evaluate the effects of orthogonality and dynamic budget adaptation using T5-large.

| Method | Dataset | | |
|---|---|---|---|
| | Standard | Long | SuperNI |
| OA-Adapter | 76.0 | 69.2 | 29.3 |
| - Orthogonality | 57.1 | 52.8 | 13.7 |
| - Budget Adaptation | 73.3 | 65.6 | 24.2 |

Table 3: Comparisons of parameter efficiency between OA-Adapter and O-LoRA using T5-large.

| Method | Budget | Params | AA |
|---|---|---|---|
| O-LoRA | 16→16 | 4.72M | 75.3 |
| OA-Adapter | 16→9.95 | 1.96M-58.5% | 76.0+0.7 |
| O-LoRA | 8→8 | 2.36M | 74.5 |
| OA-Adapter | 8→6.05 | 1.18M-50.0% | 74.7+0.2 |
| O-LoRA | 4→4 | 1.18M | 73.8 |
| OA-Adapter | 4→3.18 | 0.63M-46.6% | 74.1+0.3 |

intrinsic rank for O-LoRA. Remarkably, OA-Adapter achieves superior performance while using 46.6% to 58.5% fewer parameters compared to O-LoRA's fixed budget approach. Moreover, OA-Adapter maintains consistent performance excellence across all initial budget settings, demonstrating its robust dynamic allocation capabilities. These results highlight OA-Adapter's ability to effectively adapt parameter budget allocation, providing substantial efficiency benefits over static parameter budget allocation approaches.

**Heterogeneous Budget Requirements Across Tasks and Layers.** Intuitively, adapting to individual downstream datasets requires varying parameter budgets across different tasks and layers. To validate this, we analyze budget allocation patterns in CL scenarios, as shown in Figure 2. Our results reveal heterogeneous budget requirements across tasks and layer positions, confirming that optimal parameter allocation cannot follow uniform rules but demands task-specific consideration. Notably, in CL scenarios, the parameter matrices for the initial task exhibit significantly higher sparsity compared to subsequent tasks. This pattern supports our hypothesis that initial tasks primarily leverage capabilities inherent in the pretrained model, while later tasks must additionally preserve knowledge from preceding tasks, necessitating more complex parameter spaces. Comprehensive analysis is provided in Appendix A.4. These findings validate the necessity of adaptive budget allocation for CL based on the characteristics of layer, task and training sequence. Furthermore, this adaptive allocation aligns with established findings in the intrinsic dimensionality literature, which demonstrate that task complexity is positively correlated with the minimal intrinsic dimension required for successful optimization (Li et al., 2018; Gressmann et al., 2020; Aghajanyan et al., 2021). For instance, more challenging objectives typically exhibit higher intrinsic dimensions, necessitating greater parameter capacity (Li et al., 2018). In our experiments, we observe consistent empirical patterns that support this relationship: simpler topic classification tasks (e.g., AG News and DBpedia) systematically converge to lower average final ranks than more nuanced sentiment analysis tasks (e.g., Amazon Reviews and Yelp Reviews), as evidenced across multiple task orders in Figure 2 and Appendix A.4. This alignment provides additional quantitative grounding for the intuition that harder tasks receive more parameters, reinforcing the effectiveness of OA-Adapter's dynamic budget adaptation mechanism.

### 4.3 Discussions and Analysis

**Occurrence and Mitigation of Catastrophic Forgetting.** As shown in Figure 3, we demonstrate the occurrence and mitigation of catastrophic forgetting on the Standard CL Benchmark. Solid and dashed lines denote models with and without orthogonal parameter-subspace constraints, and each color corresponds to a distinct task. Shaded background bands mark the training phase of each task, and the X-axis is scaled proportionally to training steps. We observe a significant decline of each task after their training phase without orthogonal constraints. Performance on Task 2 drops to nearly zero by the end of the subsequent tasks, while Task 1 and Task 3 also decrease substantially. In contrast, the performance with orthogonal constraints remains largely maintained through Task 4, with the most severe degradation limited to only 14% performance loss on Task 2. These results demonstrate that severe forgetting phenomena occur during multitask training and orthogonal parameter subspace constraints can effectively mitigate it. Similar trends are consistently observed under other task orders, as detailed in Appendix A.5. Additionally, as expected, each task achieves higher performance more rapidly during its corresponding training phase without orthogonal

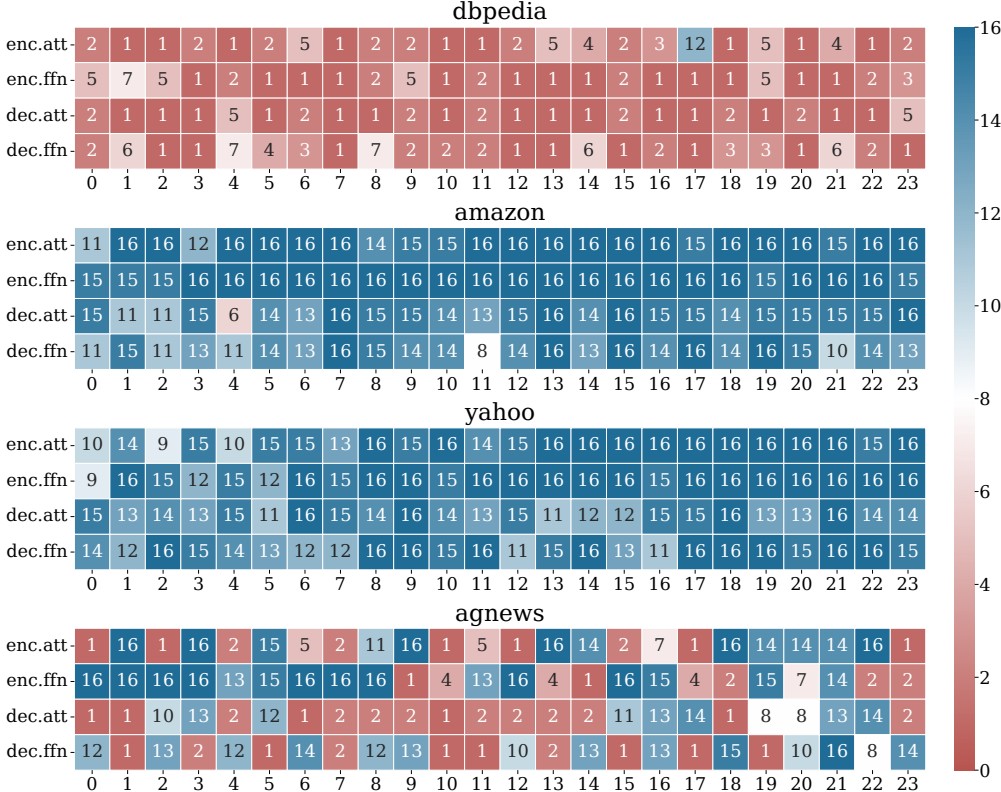

Figure 2: Final dimensions after sequential training following Order-1 using T5-large with OA-Adapter across four datasets (i.e., DBpedia, Amazon, Yahoo, AG News). The X-axis is the index of T5-large layers, and the Y-axis indicates different layers OA-Adapter applies to.

constraints, though final performance is not substantially higher than with orthogonal constraints. This suggests that the model has greater flexibility in parameter update directions when not constrained to preserve previous knowledge, and orthogonal subspace still maintain sufficient capacity for subsequent tasks. Interestingly, tasks exhibit brief performance recovery at the beginning of subsequent training phases before further decline. Performance on Task 1 recovers to nearly 100% at the start of Task 3 and to approximately 60% during Task 4. Although Task 2 drops to near-zero by the end of Task 3, it improves rapidly early in Task 4 training. This behavior resembles human memory, where disused knowledge can be reactivated with minimal effort. This suggests that catastrophic forgetting may only superficially mask a latent capacity, and knowledge representations remain partially preserved despite significant performance degradation.

**Mitigating Capacity Saturation in Late-Stage Tasks.** A critical challenge in long-sequence continual learning is the saturation of parameter space. Fixed-budget methods like O-LoRA impose rigid constraints (e.g., fixed rank or strict orthogonality within a limited subspace), which can exhaust the model's learnable capacity as the task sequence lengthens. This often results in suboptimal performance for tasks encountered later in the sequence. To verify the effectiveness of our adaptive budget mechanism in mitigating this issue, we analyze the performance of the final five tasks in the Long Sequence Benchmark (Tasks 11-15) using the T5-Large model following Order 4. We compare OA-Adapter against O-LoRA and Single-Task Learning (STL), which serves as the theoretical upper bound. As illustrated in Table 4, O-LoRA exhibits a noticeable performance gap compared to STL on these late-stage tasks, suggesting that the remaining parameter space is insufficient to capture the specific features of these new tasks effectively. In contrast, OA-Adapter dynamically allocates bottleneck dimensions based on task difficulty. Consequently, it achieves performance nearly identical to the STL upper bound. This result confirms that our adaptive approach

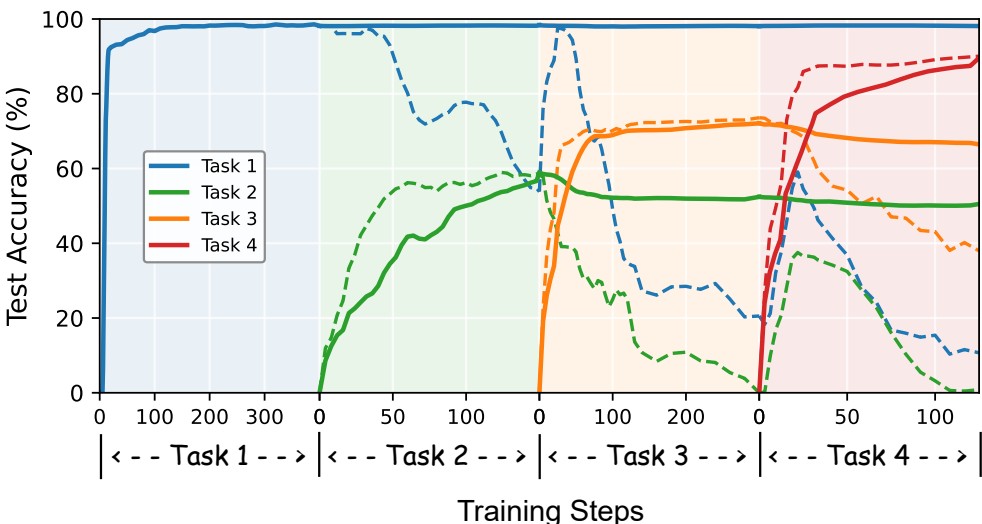

Figure 3: Occurrence and mitigation of catastrophic forgetting during sequential training following Order-1 using T5-large across multiple tasks.

Table 4: Performance comparison on the final 5 tasks of the Long Sequence Benchmark Order 4. Small red values indicate the performance gap relative to Single-Task Learning (STL).

| Method | Task 11 | Task 12 | Task 13 | Task 14 | Task 15 |
|---|---|---|---|---|---|
| STL | 92.4 | 94.8 | 86.1 | 74.3 | 62.2 |
| OA-Adapter | 92.1-0.3 | 94.5-0.3 | 85.8-0.3 | 73.8-0.5 | 61.5-0.7 |
| O-LoRA | 86.5-5.9 | 91.0-3.8 | 79.2-6.9 | 65.4-8.9 | 54.1-8.1 |

prevents capacity saturation, ensuring that the model retains sufficient plasticity to learn new, complex tasks even at the end of a long training sequence.

**Budget Adaptation Mechanism Analysis.** To assess the impact of our threshold strategy, we compare two policies: (a) a fixed, non-learnable threshold and (b) our proposed learnable threshold that adapts across different layers and tasks during training, as described in Section 3.2. These strategies are assessed using the T5-large model across three task orders on the Standard CL Benchmark. The results, depicted in Table 5, reveal that the dynamic threshold consistently demonstrates superior performance compared to the fixed threshold. This confirms that the dynamic threshold mechanism enables bidirectional adjustment of budget allocation, thereby enhancing more flexibility in the optimization process.

**Pre-trained Model Analysis.** We investigate the performance of OA-Adapter and O-LoRA across varying models (T5-base, T5-large, T5-XL, Qwen3-4B, Llama-7B) using the Standard CL Benchmark. The results, depicted in Table 6, reveal that OA-Adapter's average accuracy consistently improves, which suggests that OA-Adapter effectively leverages the increased representational capacity of larger models. The consistent superiority across different models indicates that OA-Adapter's mechanism provides effective protection against catastrophic forgetting while enabling precise task-specific optimization, regardless of the underlying model architecture's complexity and scales.

**Sensitivity to Orthogonality Coefficient.** To evaluate the impact of the orthogonality regularization strength on continual learning performance, we conducted a sensitivity analysis on the coefficient $\lambda_{\text{orth}}$ using the T5-Large model across three different task orders on the Standard CL Benchmark. As shown in Table 7, the optimal regularization strength varies by task order (e.g., $\lambda_{\text{orth}} = 5$ for Order 1 and $\lambda_{\text{orth}} = 1$ for Order

Table 5: Comparisons of threshold strategies.

| Threshold | | Order | | | |
|---|---|---|---|---|---|
| Initial Value | Strategy | 1 | 2 | 3 | avg |
| 1e-3 | fixed | 71.5 | 71.1 | 71.4 | 71.3 |
| | dynamic | **73.0** | **76.4** | **74.6** | **74.7** |
| 1e-4 | fixed | 73.7 | 73.1 | 71.4 | 72.7 |
| | dynamic | **75.7** | **75.6** | **75.1** | **75.5** |
| 1e-5 | fixed | 72.8 | 73.5 | 72.3 | 72.9 |
| | dynamic | **74.8** | **74.9** | **73.9** | **74.5** |

Table 6: Comparisons of pre-trained models.

| Model | Method | Order | | | |
|---|---|---|---|---|---|
| | | 1 | 2 | 3 | avg |
| T5-base | O-LoRA | 73.9 | **74.8** | 74.1 | 74.3 |
| | OA-Adapter | **74.7** | **74.8** | **74.5** | **74.7** |
| T5-large | O-LoRA | 75.0 | 75.4 | 75.6 | 75.3 |
| | OA-Adapter | **75.7** | **76.2** | **76.1** | **76.0** |
| T5-XL | O-LoRA | 77.9 | **78.5** | 77.4 | 77.9 |
| | OA-Adapter | **78.0** | 78.2 | **77.9** | **78.0** |
| Qwen3-4B | O-LoRA | 76.0 | 75.7 | 75.3 | 75.7 |
| | OA-Adapter | **76.7** | **76.4** | **77.1** | **76.7** |
| Llama-7B | O-LoRA | 76.2 | 76.3 | 75.7 | 76.1 |
| | OA-Adapter | **76.4** | **76.8** | **76.7** | **76.6** |

Table 7: Effect of the orthogonality coefficient $\lambda_{\text{orth}}$.

| $\lambda_{\text{orth}}$ | Order 1 | Order 2 | Order 3 | Average |
|---|---|---|---|---|
| 0.5 | 74.3 | 75.3 | 74.5 | 74.7 |
| 1 | 73.4 | 74.8 | **75.1** | 74.4 |
| 2 | 73.5 | 74.9 | 73.2 | 73.9 |
| 3 | 74.2 | 73.9 | 74.1 | 74.1 |
| 4 | 74.7 | **76.4** | 73.6 | 74.9 |
| 5 | **75.8** | 75.6 | 74.6 | **75.3** |

Table 8: Compatibility with replay-based strategies.

| Method | AA↑ | F.Ra↓ | BWT↑ |
|---|---|---|---|
| Replay | 20.5 | 14.7 | -15.8 |
| SAPT − Replay | 11.1 | 42.8 | -40.4 |
| SAPT + Replay | 44.2 | 7.7 | -6.79 |
| SAPT + ARM | 50.8 | 1.2 | -1.04 |
| OA-Adapter | 29.3 | 8.2 | -6.0 |
| OA-Adapter + Replay | 48.3 | 1.2 | -0.87 |

3), reflecting a context-dependent plasticity-stability trade-off. Therefore, unless otherwise specified, the reported main results use the $\lambda_{\text{orth}}$ selected by the best validation AA averaged across orders. This protocol ensures that the regularization strength is adaptively tailored to the specific requirements of the learning trajectory, yielding stable performance across long sequences with distribution shifts.

**Compatibility with Replay and Comparison with Replay-based Method SAPT.** To further demonstrate the flexibility of OA-Adapter, we investigate its compatibility with replay-based continual learning strategies. Replay-based methods remain a prominent approach for mitigating catastrophic forgetting despite their practical drawbacks. Specifically, we combine OA-Adapter with a simple experience replay buffer and compare the resulting hybrid against SAPT (Zhao et al., 2024), a strong replay-based baseline, on the challenging SuperNI benchmark using the T5-large model. As shown in Table 8, the OA-Adapter+Replay hybrid achieves an average accuracy of 48.3%, substantially outperforming SAPT+Replay (44.2%) while maintaining low forgetting. This consistent gain highlights that OA-Adapter integrates effectively with replay mechanisms, leveraging stored exemplars to further stabilize learning without compromising its adaptive parameter allocation. While the OA-Adapter+Replay hybrid achieves slightly lower performance than the reported SAPT+ARM (50.8%), this gap is expected given that ARM is a specialized replay mechanism tailored specifically to SAPT's attention-based architecture (Zhao et al., 2024). ARM introduces additional complexity and computational overhead through its adversarial memory management and targeted replay strategy, which are tightly coupled with SAPT's design. In contrast, our hybrid employs a simple, generic experience replay buffer that is agnostic to the underlying parameter allocation mechanism. Despite this simplicity, OA-Adapter+Replay delivers highly competitive results, demonstrating near-state-of-the-art performance with significantly lower implementation complexity. Nevertheless, replay-based methods—whether standard experience replay or specialized variants like ARM—incur substantial practical costs, including large-scale storage of task-specific data (raising privacy and memory concerns) or computationally intensive synthetic data generation. These limitations restrict their scalability in real-world, resource-constrained, or privacy-sensitive applications. Given that replay-based mitigation is not the primary focus of this work, and considering the associated overheads, we elected to center our core contributions on a fully replay-free,

memory-efficient paradigm that achieves strong performance through adaptive parameter allocation alone. Exploring tighter integration with advanced replay techniques remains a promising direction for future work, where additional resources are available.

## 5 Conclusion

In this paper, we introduce OA-Adapter, a novel parameter-efficient approach for continual learning (CL) in LLMs that considers both dynamic budget adaptation and orthogonal subspace learning in an end-to-end training stage. Comprehensive experiments demonstrate OA-Adapter's consistent superiority over existing methods across multiple benchmarks while using significantly fewer parameters. The observed heterogeneity in optimal parameter allocation across tasks and layers validates the necessity of our budget-adaptive approach. As the first work to integrate budget adaptation into parameter-efficient fine-tuning for CL in LLMs, OA-Adapter establishes a new paradigm that jointly optimizes parameter budget allocation and knowledge preservation. This advancement paves the way for efficient and effective adaptation of LLMs to evolving real-world.

### Acknowledgments

This work was supported by the National Key Research and Development Program of China under Grant 2023YFB4301900 and Beijing Nova Program 2024102. The work of L. Li was supported in part by the Natural Science Foundation of Guangdong Province under Grant 2026A1515011679, the Basic and Frontier Research Project of PCL under Grant 2025QYB014. The work of M. Pan was supported in part by the US National Science Foundation under grants CNS-2107057, CNS-2318664, CSR-2403249, and CNS-2431596.

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

# A    Appendix

## A.1    LLM Usage Declaration

Large language models (e.g., ChatGPT) were used solely for language editing and formatting. They did not contribute to the conception, design, implementation, analysis, data generation or labeling, or evaluation of the methods and results. All technical content and claims were authored and verified by the authors, and no personal, proprietary, or sensitive data were shared with LLM services.

## A.2    Datasets and Task Orders.

Table 9 and  10 show details of the datasets we used for our experiments, along with their evaluation metrics. Overall, we used datasets from the Standard CL benchmark Zhang et al. (2015), GLUE Wang et al. (2019b), and SuperGLUE Wang et al. (2019a) benchmarks, and added the IMDB movie reviews dataset Maas et al. (2011), following Razdaibiedina et al. (2023). Additionally, we incorporated the SuperNI Benchmark Wang et al. (2022a), a benchmark of diverse NLP tasks with expert-written instructions, for more rigorous evaluation of our model. Furthermore, Table 11 shows details of task orders used in our CL experiments.

Table 9: The details of 15 classification datasets in the Long Sequence Benchmark. NLI denotes natural language inference, QA denotes questions and answers task. First five tasks correspond to the Standard CL Benchmark, all other tasks are used in long-sequence experiments.

| Dataset | Category | Task | Domain | Metric |
|---------|----------|------|--------|--------|
| Yelp | CL Benchmark | Sentiment Analysis | Yelp Reviews | Accuracy |
| Amazon | CL Benchmark | Sentiment Analysis | Amazon Reviews | Accuracy |
| DBPedia | CL Benchmark | Topic Classification | Wikipedia | Accuracy |
| Yahoo | CL Benchmark | Topic Classification | Yahoo Q&A | Accuracy |
| AG News | CL Benchmark | Topic Classification | News | Accuracy |
| MNLI | GLUE | NLI | Various | Accuracy |
| QQP | GLUE | Paraphrase Detection | Quora | Accuracy |
| RTE | GLUE | NLI | News, Wikipedia | Accuracy |
| SST-2 | GLUE | Sentiment Analysis | Movie Reviews | Accuracy |
| WIC | SuperGLUE | Word Sense Disambiguation | Lexical Databases | Accuracy |
| CB | SuperGLUE | NLI | Various | Accuracy |
| COPA | SuperGLUE | QA | Blogs, Encyclopedia | Accuracy |
| BoolQA | SuperGLUE | Boolean QA | Wikipedia | Accuracy |
| MultiRC | SuperGLUE | QA | Various | Accuracy |
| IMDB | SuperGLUE | Sentiment Analysis | Movie Reviews | Accuracy |

## A.3    Task Instructions

Consistent with the experimental settings of previous works (Wang et al., 2023a; Zhao et al., 2024), we used instruction tuning in the Stand CL Benchmark and Long Sequence Benchmark experiments. Detailed instructions used for instruction tuning across different tasks are provided in Table 12. This approach offers dual advantages: it incorporates human expertise for efficient learning while enabling models to better capture underlying principles through explicit guidance, thereby enhancing generalization capabilities.

Table 10: The details of 15 datasets in the SuperNI Benchmark.

| Dataset name | Task | Metric |
|---|---|---|
| task639_multi_woz_user_utterance_generation | dialogue generation | Rouge-L |
| task1590_diplomacy_text_generation | dialogue generation | Rouge-L |
| task1729_personachat_generate_next | dialogue generation | Rouge-L |
| task181_outcome_extraction | information extraction | Rouge-L |
| task748_glucose_reverse_cause_event_detection | information extraction | Rouge-L |
| task1510_evalution_relation_extraction | information extraction | Rouge-L |
| task002_quoref_answer_generation | question answering | Rouge-L |
| task073_commonsenseqa_answer_generation | question answering | Rouge-L |
| task591_sciq_answer_generation | question answering | Rouge-L |
| task511_reddit_tifu_long_text_summarization | summarization | Rouge-L |
| task1290_xsum_summarization | summarization | Rouge-L |
| task1572_samsum_summary | summarization | Rouge-L |
| task363_sst2_polarity_classification | sentiment analysis | accuracy |
| task875_emotion_classification | sentiment analysis | accuracy |
| task1687_sentiment140_classification | sentiment analysis | accuracy |

Table 11: Eight different task sequence orders utilized in our experiments. Orders 1-3 follow the Standard CL Benchmark as established by previous research, focusing on a more traditional task sequence. Orders 4-6 are customized for long-sequence experimentation, encompassing 15 tasks each and are structured according to the methodologies outlined in Razdaibiedina et al. (2023). Orders 7-8 correspond to the SuperNI benchmark.

| Benchmarks | Order | Task Sequence |
|---|---|---|
| Standard CL Benchmark | 1 | dbpedia → amazon → yahoo → ag |
| | 2 | dbpedia → amazon → ag → yahoo |
| | 3 | yahoo → amazon → ag → dbpedia |
| Long Sequence Benchmark | 4 | mnli → cb → wic → copa → qqp → boolqa → rte → imdb → yelp → amazon → sst-2 → dbpedia → ag → multirc → yahoo |
| | 5 | multirc → boolqa → wic → mnli → cb → copa → qqp → rte → imdb → sst-2 → yelp → amazon → ag → dbpedia → yahoo |
| | 6 | yelp → amazon → mnli → cb → copa → qqp → rte → imdb → sst-2 → dbpedia → ag → yahoo → multirc → boolqa → wic |
| SuperNI Benchmark | 7 | task1572 → task363 → task1290 → task181 → task002 → task1510 → task639 → task1729 → task073 → task1590 → task748 → task511 → task591 → task1687 → task875 |
| | 8 | task748 → task073 → task1590 → task639 → task1572 → task1687 → task591 → task363 → task1510 → task1729 → task181 → task511 → task002 → task1290 → task875 |

The consistent instruction-based framework allows for direct performance comparisons while leveraging the benefits of natural language supervision.

Table 12: Instructions for different tasks used in our experiments.

| Task | Prompts |
|---|---|
| NLI | What is the logical relationship between the "sentence 1" and the "sentence 2"? Choose one from the option. |
| QQP | Whether the "first sentence" and the "second sentence" have the same meaning? Choose one from the option. |
| SC | What is the sentiment of the following paragraph? Choose one from the option. |
| TC | What is the topic of the following paragraph? Choose one from the option. |
| BoolQA | According to the following passage, is the question true or false? Choose one from the option. |
| MultiRC | According to the following passage and question, is the candidate answer true or false? Choose one from the option. |
| WiC | Given a word and two sentences, whether the word is used with the same sense in both sentence? Choose one from the option. |

### A.4 Heterogeneous Budget Requirements Across Tasks and Layers.

As discussed in Section 4.2, we extend our analysis to investigate budget allocation in the context of continual learning. Here, we further present results under other task orders, as illustrated in Figure 4 and 5. These findings corroborate our analysis in Section 4.2: (a) The relationship between performance and parameter budget does not follow constant rules but rather necessitates case-specific consideration. (b) Within continual learning scenarios, the first task primarily focuses on acquiring capabilities built upon the pretrained model, whereas subsequent tasks must additionally preserve knowledge from preceding tasks, thus requiring more nuanced fine-tuning. This further substantiates our analysis that different tasks in the CL scenarios require varying budgets, and that allocating budgets according to training sequence and task characteristics is both necessary and justified.

### A.5 Occurrence and Mitigation of Catastrophic forgetting.

To further validate the effectiveness and consistency of orthogonal parameter subspace constraints, we conduct sequential training following Order-2 and Order-3, with results illustrated in Figure 6 and 7, respectively. Consistent with the results reported in Section 4.3, we observe severe catastrophic forgetting in the absence of orthogonal constraints, especially for earlier tasks. In contrast, models trained with orthogonal parameter subspace constraints are able to preserve performance across all tasks to a much greater extent. Notably, although the specific tasks affected most by forgetting vary depending on the task order, the general trend holds: orthogonal constraints provide consistent mitigation of forgetting regardless of task permutation. These results reinforce our earlier findings and highlight the robustness of orthogonal subspace regularization as a general mechanism for alleviating forgetting in continual learning scenarios.

### A.6 Compute and Memory Cost Analysis

**Training cost and GPU memory.** We benchmark the training cost of OA-Adapter against O-LoRA under Task Order 1 on the Standard CL Benchmark. OA-Adapter demonstrated slightly lower GPU memory usage, averaging 68.9% compared to 70.4% for O-LoRA. OA-Adapter achieved a significantly faster average

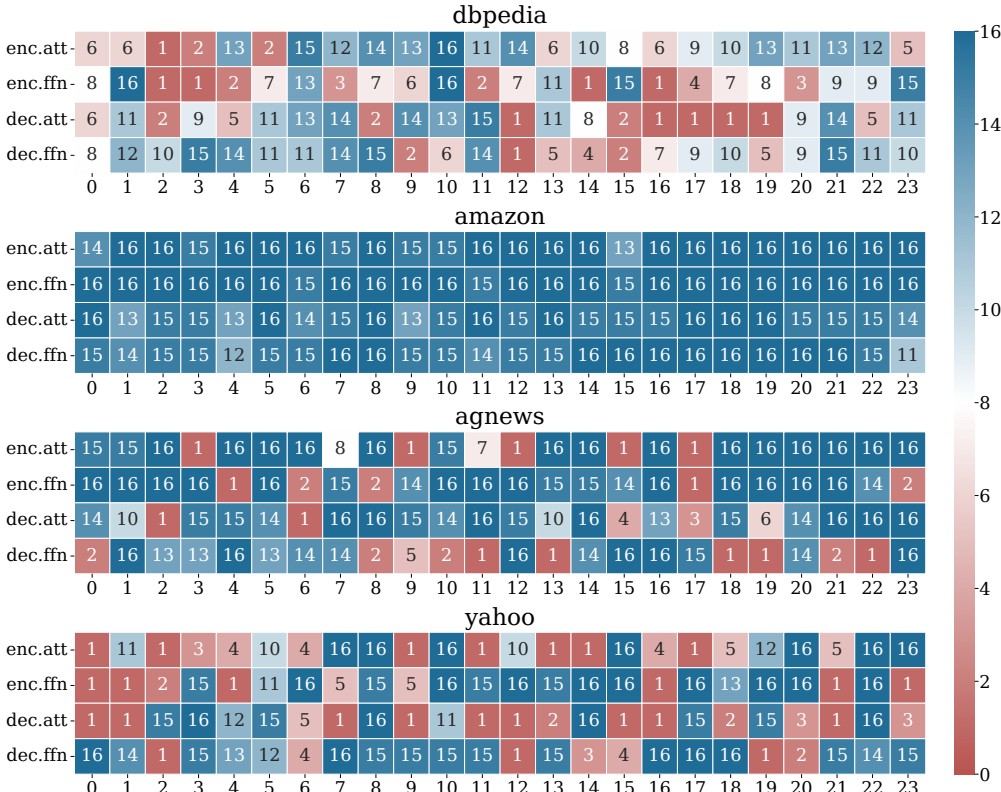

Figure 4: Final dimensions after sequential training following Order-2 with OA-Adapter.

training time per step: 0.357 seconds versus 0.582 seconds for O-LoRA, suggesting a clear advantage in training efficiency.

**Overhead from orthogonality regularization.** We further isolate the per-step time consumed by orthogonality computations for OA-Adapter (bottleneck dimension 24) on the Standard CL Benchmark, as shown in Table 13. The per-step overhead grows with the number of previously seen tasks, resulting in an average of $0.0179\,\mathrm{s}$ per step across the first four tasks, i.e., a 4.6% relative overhead compared to the total per-step time of $0.357\,\mathrm{s}$.

Table 13: Per-step time spent on orthogonality computations for OA-Adapter.

| Task | 1 | 2 | 3 | 4 | avg |
|---|---|---|---|---|---|
| Time (s) | 0.0061 | 0.0130 | 0.0199 | 0.0269 | 0.0179 |

## A.7 Limitations and Further Research Directions.

While our method has demonstrated effectiveness in empirical evaluations, there are a few limitations and potential directions of research to consider. Firstly, although our method does not rely on task identification during inference, it still requires task identification during training. Exploring methods for task-agnostic training would be a valuable future direction. Additionally, while our findings in Figure 3 reveal that models do not completely forget knowledge from previous tasks due to cross-task interference, the underlying mechanisms and potential ways to leverage this phenomenon remain unexplored in our current approach. This represents a promising new perspective and direction for future continual learning research that warrants further investigation.

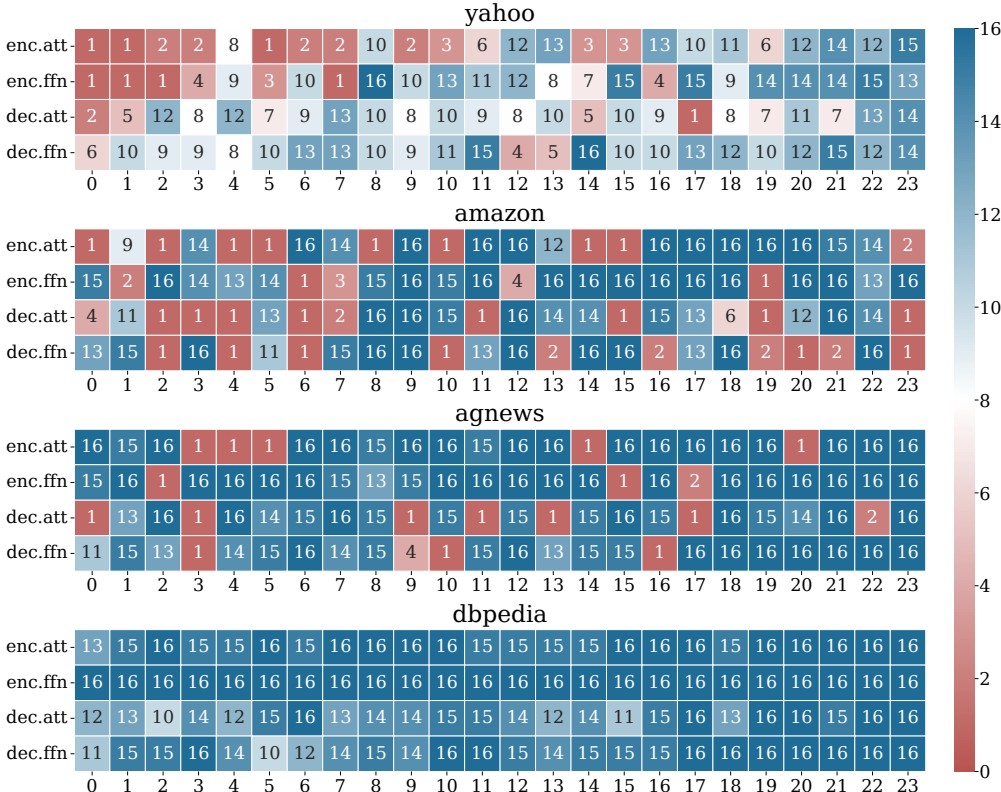

Figure 5: Final dimensions after sequential training following Order-3 with OA-Adapter.

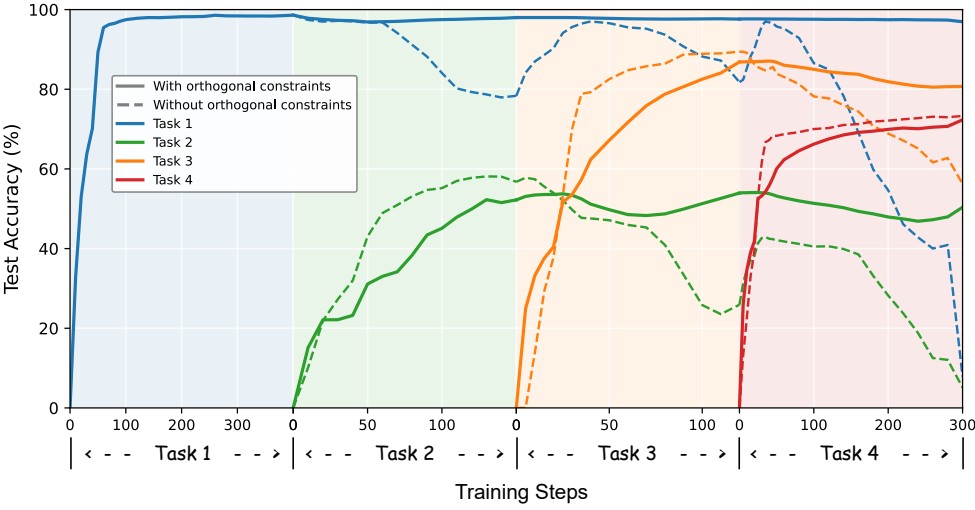

Figure 6: Occurrence and mitigation of catastrophic forgetting during sequential training following Order-2 across multiple tasks.

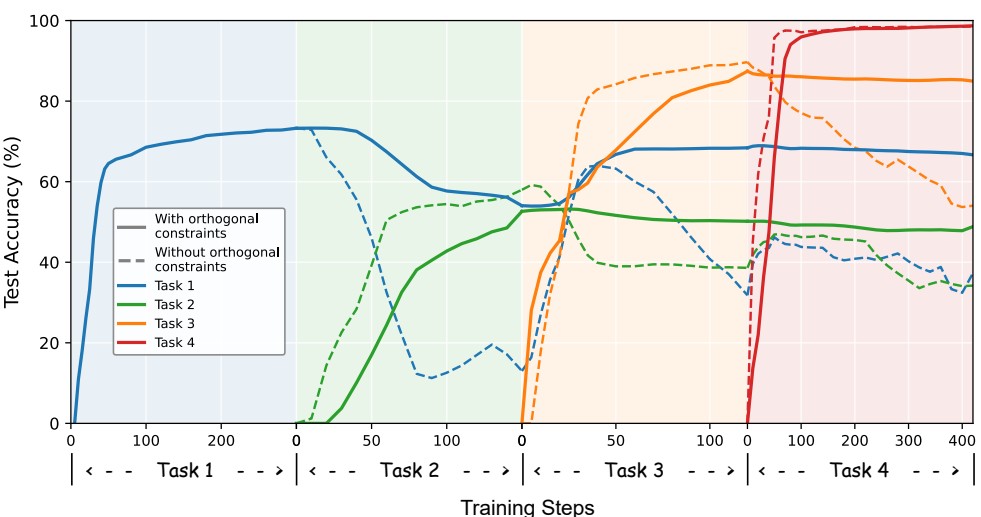

Figure 7: Occurrence and mitigation of catastrophic forgetting during sequential training following Order-3 across multiple tasks.

