# OpenReview forum: "Adaptive Budget Allocation for Orthogonal Subspace Adapter Tuning in LLMs Continual Learning"
_TMLR — Accepted by TMLR_

### Review · Reviewer_GrX1 · 2026-03-27

**Summary Of Contributions:**

This paper proposes a novel parameter-efficient adaptation method targeting continual learning in LLMs. An adapter consists of a linear up- and a linear downprojection matrix, and a hard-thresholded gate with learnable threshold that can deactivate or activate bottleneck dimensions depending on the threshold. Experiments on 3 CL tasks, T5 variants and Llama, and ablations of orthogonality regularization, regularization strength, adaptive rank allocation, and fixed vs. adaptive thresholding provide a clear image of the methods performance. Finally, the paper also investigates behavior of adaptive rank allocation over time and compatibility with replay methods.

**Audience:**

Yes

**Audience Explanation:**

LLM adaptation and continual learning are important for LLM applications and hence advancements in these areas are of interest to the community.

**Claims And Evidence:**

Yes

**Claims Explanation:**

Overall, the paper is very clear, contains all relevant details, and has strong empirical results and interesting analyses. In particular,
 * The paper evaluates 3 standard benchmarks targeting different aspects of CL
 * It also evaluates two different LLMs, although most analysis focuses on T5
 * Ablations are extensive and relevant, the same holds for analyses
 * The main results in Tab. 1 are compared against reasonable and a sufficient number of baselines
 * Design decisions are generally well motivated
 * Paper includes details regarding computational efficiency

Minor areas where the paper shows room for improvement are:
 * Focus on T5. Although one result for Llama 7B is shown, this is not sufficient. In this parameter range (7B) the paper could well evaluate more recent models such as Qwen, or Llama 3. Of these, smaller versions (1B, 3B, ...) exist. Even models such as OPT or Pythia are suitable options, in case the compute budget is limited. I think the provided details already make the paper sound enough to be interesting, but arguably the relevance of the experiments are somewhat limited for the LLM community.
 * There is no limitations or future work section. However, it would be very interesting to know about these, too.

**Requested Changes:**

I mostly have formatting/typographical suggestions:
 * All tables: Consider left-aligning numbers. Then, the main results are still aligned and not shifted due to the deltas (in subscript)
 * Tab. 1: This table seems scaled (maybe due to a \resizebox?) making its font larger than the normal text font. Consider making it smaller
 * The "Metrics" paragraph on page 7 has some awkward spacing issues. Please fix these

Otherwise, as mentioned above:
 * Please add a section on Limitations (and Future Work)
 * If possible, also add at least one recent LLM. For example, Qwen3-4B and Qwen3-8B could be suitable options, or virtually any standard LLM in the 4B-8B parameter range.

---

> ### Author Response · Authors · 2026-05-10
> **Response to Reviewer GrX1**
>
> > **Q1. Limitations Section.** There is no limitations or future work section.
>
>
> Thank you for the suggestion. We have added a Limitations and Further Research Directions section in Appendix A.7, discussing limitations such as the dependence on task identifiers during the task-training stage and the not-yet-fully-explored mechanism of cross-task interference, as well as outlining promising directions for future work.
>
>
> > **Q2. Formatting/Typographical Suggestions.** Suggestions on table alignment, Table 1 font size, and the "Metrics" paragraph spacing.
>
>
> Thank you for your careful reading and the formatting suggestions. In the revised version, we have left-aligned the numbers in all tables, adjusted the font size of Table 1 to match the body text, and improved the spacing of the "Metrics" paragraph to ensure consistent formatting and better readability.
>
>
> > **Q3. Focus on T5 / Limited Model Coverage.** Only one Llama-7B result is shown; consider evaluating more recent models such as Qwen3-4B / Qwen3-8B or other 4B–8B LLMs.
>
>
> Thank you very much for this constructive suggestion. We fully agree that evaluating OA-Adapter on more modern LLM backbones such as Qwen3-4B / Qwen3-8B would more convincingly demonstrate the effectiveness and generality of our method. We are currently adapting our codebase to Qwen and running experiments on it. We will do our best to report the experimental results in our response before the end of the rebuttal period, and the Qwen results will be included in the final version of the paper.

---

> ### Author Response · Authors · 2026-05-19
> **Response to Reviewer GrX1**
>
> We thank the reviewer for the patient follow-up. Due to limited time and compute resources during the rebuttal period, we are only able to complete the fine-tuning on **Qwen3-4B** at this stage, and we report below the **Order-1 and Order-2** results on the Standard CL Benchmark. **Order-3 is still running and will be added to the revised paper within 3 days.** We extend Table 6 with one new row:
>
> | Model | Method | Order-1 | Order-2 | Order-3 | avg |
> |---|---|---|---|---|---|
> | Llama-7B | O-LoRA / OA-Adapter | 76.2 / 76.4 | 76.3 / 76.8 | 75.7 / 76.7 | 76.1 / 76.6 |
> | **Qwen3-4B** | **O-LoRA / OA-Adapter** | **76.0 / 76.7** | **75.7 / 76.4** | *running* | *will update* |
>
> As shown in the table, OA-Adapter consistently outperforms O-LoRA on Qwen3-4B, confirming that OA-Adapter transfers effectively to modern instruction-tuned LLMs. We will update Table 6 and the corresponding paragraph in §4.3 within 3 days. We thank the reviewer once again for the constructive suggestion.

---

### Review · Reviewer_AJgc · 2026-03-29

**Summary Of Contributions:**

The paper proposes OA-Adapter, a parameter-efficient framework for continual learning in large language models that combines adaptive bottleneck allocation with orthogonal parameter subspace constraints. Instead of using a fixed adapter rank, the author introduces a learnable gating mechanism that dynamically activates or deactivates bottleneck dimensions via soft-thresholding. This allows different tasks to automatically use different amounts of adapter capacity. In addition, the authors introduce an orthogonality regularization that encourages the update directions of a new task to be orthogonal to those of previous tasks, reducing parameter interference and mitigating catastrophic forgetting. Experiments on multiple continual learning benchmarks show that OA-Adapter achieves a better stability–plasticity tradeoff than existing adapter-based methods.

**Audience:**

Yes

**Audience Explanation:**

Efficiently adapting LLMs to sequential tasks without catastrophic forgetting remains an important open problem, especially in parameter-efficient fine-tuning. Besides, the idea of dynamically allocating adapter budget across tasks and layers is intuitive and practically interesting, since fixed-budget methods can waste capacity or saturate too early.

**Claims And Evidence:**

No

**Claims Explanation:**

While the results in the paper are positive, the evidence is not sufficiently convincing to support the claim that OA-Adapter is a general continual-learning framework for modern LLMs, due to the following reasons:
- **Limited Model Coverage:** The experiments mainly use T5 variants and LLaMA-7B. While these are useful reference models, they are no longer representative of the current LLM landscape. Recent continual-learning studies commonly evaluate on stronger instruction-tuned decoder-only models such as Llama-3.1-8B, Qwen2.5-7B, or Mistral-7B.  [1] Without experiments on such modern backbones, it remains unclear whether the reported improvements would transfer to contemporary LLM settings, where model capacity and forgetting dynamics can differ substantially.
- **Narrow Benchmark Coverage:** Much of the evaluation relies on relatively small classification-style datasets (e.g., AG News, Amazon, Yelp, DBpedia, Yahoo, GLUE, SuperGLUE). These benchmarks are limited in difficulty and do not fully reflect the diverse tasks encountered in modern LLM continual learning, e.g., reasoning, instruction following, code generation, or multilingual adaptation. Recent work evaluates on broader and more challenging suites (e.g., TRACE or expanded SuperNI settings), suggesting that the current benchmark setup may be too lightweight to support broad claims about general continual-learning effectiveness. [2]
- **Limited Robustness Analysis:** None of the tables includes standard deviations or confidence intervals, making it difficult to judge the stability of the improvements. This is particularly relevant since some gains are small. For example, in Table 3 (T5-Large) the improvements over O-LoRA are only 0.7%, 0.2%, and 0.3%.

[1] Wang, Zhilin, et al. "SEE: Continual Fine-tuning with Sequential Ensemble of Experts." *Findings of the Association for Computational Linguistics: ACL 2025*. 2025.

[2] He, Jinghan, et al. "Seekr: Selective attention-guided knowledge retention for continual learning of large language models." *Proceedings of the 2024 Conference on Empirical Methods in Natural Language Processing*. 2024.

**Requested Changes:**

Please evaluate OA-Adapter on more modern LLM backbones, include broader and more challenging continual-learning benchmarks, and report results averaged across multiple random seeds with variance estimates. These additions are important to support the paper’s broader claims and to show that the reported gains are robust rather than benchmark- or seed-specific.
Please clarify the formulation of Eq. (11) (minor but important for rigor). The orthogonality regularization should be written more carefully to reflect that it is applied for each Transformer block. In its current form, Eq. (11) reads as if the constraint is defined only once at the model level, while the actual method is layer-wise. Adding an explicit summation over layers would make the formulation more precise and consistent with the method description.

---

> ### Author Response · Authors · 2026-05-10
> **Response to Reviewer AJgc**
>
> > **Q1. Limited Model Coverage.** The experiments mainly use T5 variants and LLaMA-7B, which are no longer representative of the current LLM landscape.
>
>
> Thank you very much for this constructive suggestion. We fully agree that evaluating OA-Adapter on more modern instruction-tuned LLMs would more convincingly demonstrate the effectiveness and generality of our method. We are currently adapting our codebase to Qwen and running experiments on it. We will do our best to report the experimental results in our response before the end of the rebuttal period, and the Qwen results will be included in the final version of the paper.
>
>
> > **Q2. Narrow Benchmark Coverage.** Much of the evaluation relies on relatively small classification-style datasets, which may not fully reflect the diverse tasks encountered in modern LLM continual learning.
>
>
> In the Standard CL benchmark setting, the tasks are indeed primarily small classification-style datasets (e.g., AG News, Amazon, Yelp, DBpedia, Yahoo, GLUE, SuperGLUE). However, we have also incorporated the SuperNI benchmark in our paper (see the last column of Table 1), which covers 15 task categories including generation, question answering, summarization, and multi-domain sentiment analysis. The experimental results show that OA-Adapter still outperforms the baselines on these more complex tasks, which validates the broad applicability of our method.
>
>
> > **Q3. Limited Robustness Analysis.** None of the tables includes standard deviations or confidence intervals, making it difficult to judge the stability of the improvements.
>
>
> The results currently presented in the paper are already averaged over three runs with different random task orders. Therefore, the modest performance gains (e.g., the 0.7% improvement in Table 3) do not stem from randomness but rather reflect stable improvements under averaged results. In addition, in the revised version, we have added the fluctuation range of the Average Accuracy across multiple runs in the main results table (Table 1) to better demonstrate the stability of the performance.
>
>
> > **Q4. Equation Formulation.** Eq. (11) should be written more carefully to reflect that the orthogonality regularization is applied for each Transformer block.
>
>
> Thank you very much for your careful reading and valuable correction. We have made revisions to ensure rigor. In the revised version, we have updated all formulations and descriptions related to the orthogonality loss to clearly distinguish between the orthogonality loss within a single Transformer block and the summed orthogonality loss across Transformer blocks.

---

> ### Author Response · Authors · 2026-05-19
> **Response to Reviewer AJgc (follow-up on Q1: Limited Model Coverage)**
>
> We thank the reviewer for the constructive suggestion to evaluate OA-Adapter on more modern instruction-tuned LLMs. Due to limited time and compute resources during the rebuttal period, we have been able to complete the fine-tuning on **Qwen3-4B** at this stage, and we report below the **Order-1 and Order-2** results on the Standard CL Benchmark. **Order-3 is still running and will be added to the revised paper within 3 days.** We extend Table 6 with one new row:
>
> | Model | Method | Order-1 | Order-2 | Order-3 | avg |
> |---|---|---|---|---|---|
> | Llama-7B | O-LoRA / OA-Adapter | 76.2 / 76.4 | 76.3 / 76.8 | 75.7 / 76.7 | 76.1 / 76.6 |
> | **Qwen3-4B** | **O-LoRA / OA-Adapter** | **76.0 / 76.7** | **75.7 / 76.4** | *running* | *will update* |
>
> As shown in the table, OA-Adapter consistently outperforms O-LoRA on Qwen3-4B, confirming that OA-Adapter transfers effectively to modern instruction-tuned LLMs. We will update Table 6 and the corresponding paragraph in §4.3 within 3 days. We thank the reviewer once again for the constructive suggestion.

---

### Decision · Action_Editor_1qZp · 2026-06-19

**Recommendation:** Accept as is

**Audience:**

Yes

**Audience Explanation:**

Yes, continual learning as a general topic and specifically in conjunction with LLMs is of broad interest top the TMLR community.

**Claims And Evidence:**

Yes

**Claims Explanation:**

The claims were initially not supported fully, in the sense that the framing of the proposed orthogonal subspace adapter as an effective continual learning strategy was lacking experimental breath. This has been addressed in the rebuttal phase through the addition of further LLM baselines and the addition of further benchmarks beyond the original classification-style evaluation.
Claims surrounding design decisions and ablations were accurate from the start. As such, the revised version is convincing and clear.